# A Decision-Aware Ambient Assisted Living System with IoT Embedded Device for In-Home Monitoring of Older Adults

**DOI:** 10.3390/s23052673

**Published:** 2023-02-28

**Authors:** Fatemeh Ghorbani, Amirmasoud Ahmadi, Mohammad Kia, Quazi Rahman, Mehdi Delrobaei

**Affiliations:** 1Faculty of Electrical Engineering, K. N. Toosi University of Technology, Tehran 1631714191, Iran; 2Department of Telecommunication Systems, TU Berlin, 10587 Berlin, Germany; 3Max Planck Institute for Biological Intelligence, 82319 Seewiesen, Germany; 4Department of Electrical and Computer Engineering, Western University, London, ON N6A 5B9, Canada; 5Center for Research and Technology (CReaTech), Faculty of Electrical Engineering, K. N. Toosi University of Technology, Tehran 1631714191, Iran

**Keywords:** ambient assisted living, augmented reality, fuzzy decision making, indoor positioning, internet of things

## Abstract

Older adults’ independent life is compromised due to various problems, such as memory impairments and decision-making difficulties. This work initially proposes an integrated conceptual model for assisted living systems capable of providing helping means for older adults with mild memory impairments and their caregivers. The proposed model has four main components: (1) an indoor location and heading measurement unit in the local fog layer, (2) an augmented reality (AR) application to make interactions with the user, (3) an IoT-based fuzzy decision-making system to handle the direct and environmental interactions with the user, and (4) a user interface for caregivers to monitor the situation in real time and send reminders once required. Then, a preliminary proof-of-concept implementation is performed to evaluate the suggested mode’s feasibility. Functional experiments are carried out based on various factual scenarios, which validate the effectiveness of the proposed approach. The accuracy and response time of the proposed proof-of-concept system are further examined. The results suggest that implementing such a system is feasible and has the potential to promote assisted living. The suggested system has the potential to promote scalable and customizable assisted living systems to reduce the challenges of independent living for older adults.

## 1. Introduction

Recent world population projections indicate a constant increase in older adults living alone. Mild cognitive impairment (MCI) is one of the critical causes of disability and dependency among this group of people [1]. Memory difficulties may appear in healthy older adults without any associated illnesses. Thus, older adults may need to be continuously reminded of daily living activities by their families and caregivers.

The caregivers may use the best interior home design to create a pleasant surrounding with reminders in such situations. For example, they may use open boxes (e.g., keep the container’s doors open) to make objects visually more noticeable and accessible. Alternatively, they may leave cues (e.g., “Take your medication at noon” or “Do not leave home!”). Unfortunately, such approaches are not entirely practical [2].

Recently, augmented reality (AR) technology is gaining momentum as cognitive assistance to aid people suffering from cognitive and memory impairments, particularly in the early stages [3]. It has also proved its capability in assessing older adults’ cognitive impairment response to AR-based serious games [4,5]. It has been shown that serious games are a type of multimedia game with excellent potential for rehabilitation and cognitive improvement. AR-based serious games assess a person’s cognitive performance and rehabilitation progress by monitoring their interaction with real and virtual objects in an AR environment [5,6]. The ARCoach system [7], designed by Chung Yuan Christian University researchers, incorporates an external web camera and a personal computer to superimpose information on a real object. Participants with cognitive impairments employed the ARCoach to prepare their meals. The system was capable of creating alerts (sound notifications) when food items were misplaced or inexact by detecting AR tags representing food items.

In another study, the researchers introduced a MED-AR system developed by smart AR for assisting older adults with medication-taking tasks [8]. The system presented a research methodology for tracking and distributing prearranged medications for older adults in their daily living in-home situations.

In another work, the authors proposed a system that assists people with Alzheimer’s disease by employing a smartphone AR interface through audio commands and virtual objects during their pharmacological and nonpharmacological treatment [9]. Additionally, a new approach based on AR has been introduced to help caregivers design a smart home to assist people with cognitive impairments who need reminders to complete daily living activities on their own [10].

The use of tablets and smartphones has become a standard component of our everyday lives due to their ability to exchange high-speed information via Wi-Fi, Bluetooth, and 5G/4G [11]. This has led to considerable interest in mobile-phone-based healthcare systems, and various applications have been developed [12,13]. Meanwhile, the development of internet of things (IoT) applications has also drawn considerable interest, especially in health management systems [14,15]. IoT devices for chronic diseases typically include wearables and smartphone applications to help patients track their health. Mobile applications developed for smartphones, tablets, or other wearable devices (such as smartwatches) could be incorporated into IoT systems for older adults [16]. Aged care monitoring is where the IoT systems monitor end users’ health status and general well-being. Examples include ambient assisted living (AAL), active aging, therapy and entertainment, communication, and social activities. The main objectives of this group of applications are improving the quality of life of older adults and their caregivers and promoting living safely [17]. The AAL is one of the most rapidly expanding areas, which support older adults with embedded equipment [18].

There are still some challenges regarding the AAL systems’ design process, including dynamic service availability, service mapping, and user willingness [19]. The availability of AAL services is dynamic, and handling this dynamic is a considerable challenge. Moreover, the architecture of homes and users’ needs may diverge noticeably. There should be a way to introduce a structure, so that numerous devices can be connected and be turned on or off and automatically make an appropriate decision.

Therefore, there is an unmet need to understand the various perceptions of developing an integrated AAL system, which can be customized easily to support older adults, their families, or physicians. Few research works have been conducted to interrogate the technology potential for providing monitoring services and task prompting systems as an integrated model for older adults and their caregivers. Most studies have addressed each problem separately and focused on designing a monitoring system or an assistive tool for older adults.

The main goal of the proposed work is to fill this gap and provide a conceptual model based on emerging technologies, such as IoT and AR, for designing AAL systems. To the best of our knowledge, despite significant interest, there has not been much work conducted on the integration of AR, IoT, cloud-based fuzzy decision-making engines, and indoor positioning methods to study the feasibility of implementing a comprehensive proof-of-concept system.

In a cloud-based fuzzy decision-making engine, the fuzzy logic framework can be implemented on cloud service platforms to achieve intensive computation-based fuzzy inferences. Large amounts of sensor data can be analyzed with this approach because it is versatile, efficient, scalable, and economical [20,21]. Thus, all the information for the inference system could be transmitted to a server and stored in a database. In our work, this information includes indoor positioning data, caregivers’ prompts, and various embedded devices’ statuses placed in the user’s home environment.

In order to evaluate the feasibility and accuracy of the suggested model, preliminary proof-of-concept implementation was carried out, and functional experiments were conducted based on the suggested scenarios of use and required setups throughout three different experimental conditions [2]. These experiments were carried out concerning performance evaluation parameters, such as AR message response time and the decision-making engine’s accuracy and reliability, to confirm the proposed model’s effectiveness. Furthermore, we also address the challenges regarding the architectural features and requirements for adding new fuzzy rules and new embedded devices.

Here is a brief description of how the paper is organized. Section 2 of this paper describes the materials and methods that were used in this study. The results of the implementation of a particular system according to the suggested methods and setups are summarized in Section 3 of this document. The evaluation of our model based on the use scenarios will be discussed in Section 4 in comparison to other works. In addition, this section presents the current design challenges in AAL systems and highlights the solutions we developed in response to them. Section 5 of the study concludes the study and suggests directions for further research in the future.

## 2. Materials and Methods

AAL is an emerging multidisciplinary field involving wireless networks, sensors, mobile devices, and software applications for telehealth and the monitoring of personal health care [22].

In order to improve the quality of life of older adults, the AAL system gathers data about their activities, their lifestyle, health condition, and the description of their home. The decision is made by analyzing and interpreting (1) the context and lifestyle of the person, (2) the data collected and understood by the sensors, (3) the interpretation and analysis of the data on the person’s activity, and (4) the assistance provided to the person during the realization of the activity. In brief, AAL is built on the semantics of everything that surrounds the person and accompanies them as they go about their daily lives [22].

This section explains how our proposed conceptual model can be designed to be more effective in developing AAL systems based on the common scenarios in the target population we present later on. Afterward, we propose an architecture and explain the methods, including IoT-based smart devices, object-based decision-making processes, and indoor positioning systems. Finally, we describe the software requirement for the AR application as the primary user interface.

### 2.1. Conceptual Model

The proposed model addresses the current challenges in developing AAL systems. It can also be beneficial in designing scalable and customizable AR-based technologies for users with memory impairments. The user interaction within the AR environment can be rendered by AR glasses or mobile phones.

Fog computing offers end users a range of application services, such as data processing and storage. In our work, using a local fog layer allows us to process data more easily and efficiently. By monitoring local computation and processing in the edge devices, the fog layer enhances the long-term planning and continuous system improvement of the IoT system. The processed data are sent to the cloud for further analysis.

The decision-making process is based on human-expert knowledge, which includes fuzzy if-then rules. In our work, the input and output variables used in a fuzzy decision-making system are the data collected from embedded devices in the user’s environment. Through membership values, discrete values, such as Yes and No, including relay actuator status, can be represented as absolute Yes and absolute No. Other continuous values, such as distance, can also appear. A membership function, such as the Gaussian membership function [23], determines the membership value for continuous values. In more detail, based upon the previous discussion and Figure 1, this work presents a model with the following features:(a)Using the IoT protocols, all the data information, such as localization and sensor data through embedded IoT devices in the user’s home environment, could be transmitted to the server and stored in the database. Caregivers or physicians can also override the reminder and subscribe to each particular topic if needed.(b)The MQTT protocol is based on the concept of bridging, which is available in some current MQTT broker implementations (such as Mosquitto, HiveMQ, and CloudMQTT). It connects a broker A to another broker B as a standard client, subscribing to all or a subset of the topics transmitted by clients to B. Since our task is to obtain data from the embedded devices and send notifications to the user’s AR user interface, we chose the MQTT broker HiveMQ (hivemq.com). This is a free cloud broker, which allows IoT devices to be connected to the cloud. By creating functions for each variable, we were able to separate the variables into different MQTT topics.(c)Once an event occurs in a particular home location, all the pertinent data values are checked on the server. Therefore, after the decision-making process on the cloud, data values will be updated. The caregiver and the older adults receive appropriate notifications.(d)An appropriate AR message will be sent to the user when the user interacts with a particular object (for example, when the user is looking for their medication) or when the sensor and localization tag detect an event (for example, when the user enters a dangerous zone at home). These messages are defined based on the decision of the fuzzy inference and the output values. The end user puts on an indoor positioning tag to monitor the real-time location of the user and make easy interaction with the home objects. The real-time position and orientation data values also form part of the decision-making engine’s inputs. Moreover, user activity patterns can be generated and stored in the database for further analysis.(e)The AR messages are sent to the user’s AR device while interacting with selected objects, or an event is detected from the sensor and anchors data. These messages are the output of the decision-making engine included in the service-based application.

### 2.2. Sample Scenarios

The evaluation of the proposed model is based on situations where the user is not able or overlooks to carry out their daily activities. Some common difficulties of a person with memory impairment were described in this section and further evaluated in Section 3.

The following scenarios focus on an older adult with memory impairment who may cook a meal, take medication, and go outside for a walk with some level of autonomy.

The user may skip an activity, such as having a shower or eating breakfast, because they fail to recall it.The user often repeats activities because they do not remember what they already did, such as taking their medication.The user may forget to complete a task correctly or avoids a critical activity, such as turning off the stove or closing the main entrance.

Our proposed model can potentially tackle such issues by displaying relevant cues to help users follow their daily activities’ usual course of action.

### 2.3. System Architecture

The message queuing telemetry transport (MQTT) is a frequently used protocol in the application layer of an IoT structure to convey information between the embedded nodes because of its scalability and simplicity [24]. This study applies the MQTT standard, since the message header requires only 2 bytes of the data packet, making it a lightweight published-subscribed messaging protocol. To implement the proof-of-concept AAL system, we considered an MQTT broker as the server and WeMos D1 Mini (ESP8266) components for every actuator, data coordinator of the localization system, or sensor. Figure 2 shows the MQTT protocol operation in our study.

We designed the architecture with the capability of easy integration and the addition of new sensors [25]. The application for interacting with the user was designed on an Android operating system capable of running on AR glasses and smartphones. We also employed a Windows application user interface for caregivers to display the notifications associated with the sensors’ events and control the embedded devices in the home environment, such as actuators. All the transmission to the proposed AAL system is performed throughout MQTT with data values serialized in JavaScript object notation (JSON) configuration.

### 2.4. Embedded Devices

As the daily routine of older adults is dependent on their home objects in the surroundings [26], we propose six types of actuators and sensors placed in various locations in their home, including (1) a rain sensor (on the terrace), (2) flame sensor (in the kitchen), (3) gas sensor (in the kitchen and bedroom), (4) temperature sensor placed in the TV room, (5) humidity sensors in the flower pots, and (6) relay actuators to close or open the main entrance or the drawers.

The rain sensor module is used to measure the amount of precipitation in the balcony. Temperature and humidity are also monitored by DHT11 sensors. Fire is spotted by IR infrared flame sensors, and gas leakage is spotted by MQ-135 sensors placed in the kitchen and bedroom. In addition, using relay actuators with the ESP8266 is a way to close or open the main entrance or the drawers remotely. To acquire the desirable result, each module and component is interfaced with WeMos D1 mini, which has an integrated Wi-Fi module (ESP8266). Thus, the data are sent to the following WeMos D1 mini Wi-Fi module through the MQTT communication protocol.

The general model of the proposed AAL system is depicted in Figure 3. Three anchors are also required to evaluate the real-time location of the user in the home environment.

### 2.5. Object-Based Decision-Making Process

In order to update each variable on the cloud and make a proper decision based on the determined rules, we employed a fuzzy logic model. To make decisions involving imprecision and uncertainty, fuzzy logic is a way to model and make decisions using natural language [27]. This is due to its ability to handle various sources of real-world domain and data uncertainties, generating easily adaptable and explainable data-based models. In our work, we aim to take expert persons’ supervision as fuzzy if-then rules into account, so that older adults can complete their daily activities without the direct in-home supervision of caregivers or physicians. The prompting system uses fuzzy logic to decide whether prompts should be delivered based on the user’s daily activities and interactions with objects. The fuzzy logic method provides minimum computation time with the least complexity and highest accuracy based on our goal [28]. According to Ref [29], a fuzzy controller includes four sections: the inference system, rule base, defuzzifier interface, and fuzzifier interface.

Figure 4 shows the block diagram of a fuzzy control system. We consider the fuzzy technique to transform our design into an expert system, which is capable of monitoring older adults and expanding their ability to complete their daily activities.

#### 2.5.1. Fuzzy Logic Implementation

As shown in Figure 4, the fuzzy controller error signal is the difference between the output and reference signal. For instance, in our model, the distance between a specific object and the user is considered an error signal. Moreover, in our suggested system, the output variables are various kinds of multimedia messages taken by the user and the state of the actuators in the specified places, and the input variables are different types of information received from sensors and embedded devices, the real-time position and heading angle of the user. Table 1 indicates the output and input variables, membership functions, and data types. We can insert more objects located in the home and estimate the distance between them as a new fuzzy input parameter.

#### 2.5.2. Input/Output Variables

We considered each output and input variable and defined membership functions according to the fuzzy rules and data types. The principal fuzzy membership function often used to signify vague linguistic terms is a Gaussian membership function for each language expression. Thus, we defined Gaussian membership functions for variables, such as distance (Near, Far, Very Far) and heading angle (Small, Medium, Large). As linguistic variables examples, Figure 5A,B indicate the Gaussian membership functions for the distance between the specific objects and the user and their heading angle.

The output variables include multimedia, such as audio, text, and image messages. Each message can be triggered by specific input values and predefined fuzzy rules and is associated with an identification number (ID). Each message has a corresponding ID, described as a function of fuzzy singleton membership. As an instance of these variables, voice message membership functions are shown in Figure 5C.

Moreover, the state of actuators placed in the environment is defined as Boolean variables (No is the default value, and Yes means the actuator should be activated). A triangular membership function was used for this purpose.

#### 2.5.3. Fuzzy Rule Base

After determining the membership function, the rule base composed of expert if-then rules is created [30]. These rules are designed based on the common scenarios in older adults with cognitive impairments’ daily difficulties, as explained in Ref [2]. Caregivers can modify the content of multimedia messages, such as image and audio notifications. Some parts of these rules based on the situation of the user are as follows:Leaving home: If (rain status is Yes) and (distance from the main entrance is Near) and (heading angle is Small), then (image message is picture 3) and (voice message is audio 3).Cooking: If (distance from the refrigerator is Near) and (heading angle is Small), then (image message is picture 2) and (voice message is audio 2).Daily activity reminder: If (the plant’s humidity is Very dry), then (text message number is text 1).Daily activity reminder: If (distance from the TV is Near) and (heading angle is Small), then (image message is picture 6) and (voice message is audio 6).Cooking: If (time is Late Afternoon) and (distance from the oven is Near) and (heading angle is Small), then (image message is picture 4) and (voice message is audio 4).Medication reminder: If (time is Evening), then (image message is picture 5) and (voice message is audio 5).Alarm: If (flame sensor status is Yes), then (audio message is audio 7) and (relay status is Yes).Alarm: If (gas sensor status is Yes), then (text message is text 2).and (relay status is Yes).High Temperature Reminder: If (temperature sensor status is Hot), then (text message is text 3).Low Temperature Reminder: If (temperature sensor status is Cold), then (text message is text 4).Danger zone: If (distance from the fireplace is Near), then (voice message is audio 8).Danger zone: If (distance from the balcony is Near) and (heading angle is Small), then (image message is picture 9) and (voice message is audio 9).

When an event occurs (for example, the user stares at specific objects or a sensor acquires a new data value), input data values, such as position, heading angle, and sensor statuses, are then transmitted through the MQTT network protocol on the cloud. Furthermore, fuzzy rules are checked, and equivalent output parameters are then transmitted and updated on their specific topics. Thus, based on which rules are triggered, the user receives unique messages or activates a particular actuator. In our experimental tests, each of these messages was defined based on older people’s real-life scenarios. We described some of them in Section 2.2. The algorithm’s control loop refresh rate is set between 0.5 and 2 Hz to improve system performance and decrease processing time. The fuzzy engine can also scale up to more rules to improve the users’ independence.

The final point is to map a fuzzy set to a crisp set. The proposed model applies the Mamdani inference system, which provides a fuzzy set of output membership functions. There are various methods for the defuzzification process, wherein the center of gravity is the most prevalent one in the defuzzification approach; thus, we use this method.

### 2.6. User Indoor Location Identification

To monitor users’ interaction with the environment, an accurate indoor positioning system is required [31]. The user’s heading angle is also essential for estimating the object’s position. We assume that the home objects’ position is predefined; hence, the distance from the predefined objects can be easily calculated.

An inertial measurement unit (IMU) sensor (MPU-9250, InvenSense Inc., San Jose, CA, USA), including gyroscopes, accelerometers, and magnetometers, was used to estimate the user’s heading angle. Finding the heading angle comprises two challenges: compass calibration and sensor fusion. The first problem focuses on hard and soft iron calibrations and compensation [32]. The second problem is fusing the IMU data to compute Euler angles and quaternions. For solving this problem, there are generally two standard algorithms: extended Kalman filter (EKF) and magnetic, angular rate, and gravity (MARG) [33]. We implemented the EKF algorithm, which has higher accuracy.

Position estimation includes two main sections: solving the location inverse geometry problem and calculating distances from the anchors (static beacons) to the tag. To calculate the real-time distance between the specific objects and the user, first, we estimate the user’s in-home location. Our method for position estimation was developed utilizing a single-chip wireless transceiver (DW1000, Decawave, Dublin, Ireland) based on UWB technology. UWB can be used in short proximity to other radio frequency (RF) signals without generating or suffering from interference because of the variations in signal types and radio spectrum [34].

Our range-finding algorithm used in the study is a two-way-ranging, double-sided algorithm based on the time difference of arrival (TDoA) value [35]. As shown in Figure 6A, in this work, three anchors are used in the TDoA method, and the distance values can be calculated as the time it takes for the conveyed signal to travel from the anchor to the tag and back to the anchor. Figure 6B illustrates the basic idea of the round-trip time of arrival (RToA).

The RToA is utilized to evaluate the distance between each anchor and the tag (slant range). It is noted that employing a fog server enhances the efficiency of the network and decreases the bandwidth prerequisite by preparing real-time data for mobile users closer to the edge of the network [34]. In the designed local fog, based on IEEE 802.15.4 [36], estimated ranges are always conveyed between the anchors and the tag, noticed by the data coordinator node. Conclusively, in the local fog, the coordinator encapsulates and converges all the transmitted distances and broadcasts them through the MQTT network protocol to the cloud server in JSON format. Figure 7 shows the prototype used for user position estimation, the data coordinator, and the anchor used in this study.

Our study’s prototype aims to estimate the user’s indoor position. It consists of a data coordinator and three anchors. An IMU, including gyroscopes, accelerometers, and magnetometers, was used to estimate the user’s heading angle. For position estimation, we used a single-chip wireless transceiver based on ultra-wideband.

We formulated a location identification problem using ToA and RToA methods to anticipate the tag’s location with references to the anchors’ place set in the server. Each anchor propagates signals to locate the tag within an interval. The position vector $\vec{r}$ of the target concerning the reference frame position is
(1)$$\vec{r}=x_{i}+y_{j}+z_{k}=M\left(x,\;y,\;z\right)$$

The distance between each anchor and the tag can be easily calculated as
(2)$$r_{i}^{2}={\left(x_{i}-x\right)}^{2}+{\left(y_{j}-y\right)}^{2}+{\left(z_{i}-z\right)}^{2};\;i=1,\;2,\;3$$

We employed an optimization approach to solve the position formula, as this could potentially support more anchors and more dimensions, a property that makes the results more accurate. In order to figure out the optimization problem for a given cost function, we used the Nelder–Mead simplex algorithm [37] to find the user position.

### 2.7. Augmented Reality Application

The Android ARCore platform [38] (Google’s open-source AR Software Development Kit) was used to design the AR application. We defined three types of messages to interact with the user, including pictures, audio, and text. In some scenarios, the user could receive two types of messages simultaneously. Our application was developed on an Android smartphone for better visualization. We used a Samsung Galaxy S7 smartphone with a 12 MP rear camera, a quad-core Snapdragon 820 processor, and 4 GB RAM for simulation and evaluation purposes.

## 3. Results and Experimental Setup

In order to evaluate the feasibility of implementation and the accuracy of the proposed model, a proof-of-concept system was developed. Functional experiments were carried out based on the suggested scenarios and required setups throughout three experimental conditions. We created these experimental setups based on older adults’ real-life scenarios presented in Section 2. Some of these scenarios are based on defining danger zones in the user’s home; hence, the system acts to detect the danger zones. The location of the dangerous objects can be determined by the user interface application designed for the family or caregivers. Other scenarios are based on the user’s position and the sensors’ data. In this section, we present some experimental results to evaluate the reliability and accuracy of the proposed system based on the participation of a user.

### 3.1. Experiment A: Danger Zones and Reminders

To monitor the real-time location of the user by caregivers, we used the Unity cross-platform game engine because the user’s home architecture could be represented easily. Furthermore, the anchor’s location and danger zones were defined by experts. We used C# libraries to implement the proposed algorithms. In our preliminary experiments, we defined five objects with equivalent dimensions of 0.5 × 0.7 × 0.2 m^3^ placed in an 8.6 × 4.5 × 2 m^3^ laboratory environment. The orientation data were generated using an IMU embedded in the AR device to estimate the exact heading angle.

The distance between the object and the user is constantly transmitted on its topic to the MQTT message broker. The MQTT messages update once the user walks into the room, and the distance between the user and the objects changes. Therefore, the user can receive different notifications if any rules were activated by subscribing to their MQTT topic.

The experiments were repeated 20 times with different variations on the user’s localization data and IoT devices’ status. The distance between the user and the predefined object was calculated in each experiment based on the localization data. The localization data may change following the user’s movement and heading angle changes. As explained in Section 2.5.2, the Gaussian membership function for the distance between a specific object and the user is defined as {Near, Far, Very Far}, and the heading angle is defined as {Small, Medium, Large} fuzzy values.

We defined five fuzzy rules to test the system’s performance in activating appropriate messages based on the user’s interaction with the objects. These rules were designed based on daily difficulties experienced by older adults with cognitive impairments [2].

Rule 12 (medication schedule):

If the user is “Near” the drawer (Object 1) and looking at their medication (heading angle “Small”), they receive a multimedia message in which the correct medication schedule is shown.

Rule 14 (forgetting loved ones’ information):

If the user is “Near” the table (Object 2) and looking at the family picture (heading angle “Small”), they receive the name and age of the family member as a voice message.

Rule 11 (forgetting keys):

If the user is “Near” the shoe case (Object 3), they receive a voice message reminding them to take the keys if they want to leave the house.

Rule 21 (childhood diaries):

If the user is “Near” the bookcase (Object 4) and looking at their books (heading angle “Small”), they receive a voice message reminding them of their favorite book in childhood.

Rule 2 (trouble with cooking):

If the user is “Near” the stove (Object 5), they receive a multimedia message reminding them about the place of their meal.

In addition, we repeated the experiment four times for each object with different distances and heading angles to verify whether the correct rule was activated and proper messages were sent.

As summarized in Table 2, there were five specific objects, and the user’s distance from them was assessed while the user was moving. We also checked the activated fuzzy rule and its reliability in each experiment. Based on the activated rule, the output would be a specific AR message sent to the user later on. In a regular scenario, when the user is near an unsafe object (for instance, the fireplace), they receive a picture and a voice alert. This notification reminds them to stay away from the object. These activities are performed both in the game engine environment (by the designed character) and in the actual scenario (experiments). With adequate knowledge, the character’s movements can be changed in various ways to also represent the alternatives.

As discussed in Section 2, the “Near” membership function indicates the danger zone (from 1 to 5 dm). This situation activated rule number 21, as mentioned in Table 2. We also illustrated the user’s interaction while moving toward predefined objects and zones, including the living room and kitchen, as shown in Figure 8. The experimental laboratory condition was simulated on the Unity platform to monitor the indoor position of the user and their interaction with the smart objects. The home environment can be customized for each user in the game engine, and their interactions can be monitored.

A relevant experiment was performed 20 times in different conditions, as shown in Table 2 (by a single user). As a sample, Figure 8 illustrates the user’s state in experiment number 14 (Figure 8A) and experiment number 4 (Figure 8B). Based on the experiments, the accuracy of the position estimation system was measured at close to 1 dm.

As reported by Ref [2], older adults with cognitive impairments have some difficulties remembering their family’s faces and names; thus, we evaluated the system’s potential as memory assistance in another scenario. In the eighth experiment, when the user entered the room and they were looking at the family picture on the table, they received the name and age of the family member as a voice message.

Some other experiments in which the membership function was not Near (for distance) and Small (for heading angle) did not trigger any fuzzy rules. For example, in experiment number seven, the membership function of the distance was Very Far, and the membership function of the heading angle was Large. Thus, the user did not receive any message. Similar experiments can be seen in experiments 5, 7, 11, 16, 17, and 20.

In another condition, one of the variables was a member of the desired membership function and the other variable was not. Therefore, in this situation, the fuzzy rule will not be activated. As examples, please review experiments 2, 3, 4, 6, 10, 12, 13, 15, and 19 in Table 2.

Moreover, to improve the user’s daily activity, we defined other scenarios, such as reminding plant watering time based on the plant’s humidity. These scenarios are independent of the user’s movement based on the sensors’ and actuators’ data values. To this end, the humidity sensor transmits its value to the message broker, and when the humidity level drops below 40% (very dry fuzzy membership function), fuzzy rule number 3 is activated, and the user receives a text message containing plant care and watering reminder alarm.

Furthermore, we developed a Windows-based application for the family members and caregivers to remind the user about their medication time, which could be the most important event in older adults’ daily lives (they often forget the exact time and the dosage). This application allows the caregiver to set the medication time or its period, so at any time of the day, when fuzzy rule number 6 is activated, the system notifies the user by sending images and audio messages to them. The messages include information about the location and dosage of the medication.

### 3.2. Experiment B: Alerting Based on User’s Location and Sensors’ Data Values

In Section 2, the fuzzy rules were presented. In most of these rules, the user’s heading angle and distance from the objects were the primary inputs to the system. Therefore, in another experiment, we evaluated the system’s accuracy and reliability for user location identification. Figure 9 shows the pattern of the user’s movement during the first six seconds of this experiment, where the user moves toward an object (Obj1) from point 1 to point 3. The user’s location point and heading angle, anchors’ (A1, A2, and A3) position, and object’s location, according to Figure 9, are presented in Table 3.

As shown in Figure 10A, the user changed their distance relative to a specific object (Obj1) while searching the environment. As discussed in Section 2.5.2, the distance is a variable, which indicates how far the user is located from a predefined object. Our study defined a distance value between 1 and 14 dm based on our experimental space. If the distance between the user and the object is between 1 and 5 dm, we consider it “Near.” It indicates that the user is near the object and may want to make an interaction with that object. If the distance between the user and the object is between 5 and 9 dm, we consider it as “Far”; if the value is larger than 9, it will be regarded as “Very Far”. When the distance value lies within 1 to 5 dm, it becomes a member of the near fuzzy membership function. Figure 10B shows the pattern of the time variation of the user’s heading angle during this experiment at a sampling frequency of 100 Hz. However, as the heading angle changed between −12 and 12 degrees, it turned into a member of a small fuzzy membership function. As these two inputs become near and small, the system checks other input values according to the fuzzy rules. After updating the user’s location and orientation data values on the cloud, the system checks other inputs.

As shown in Figure 9, the user’s final location point was (4.8,2.8), which was near object 1′s location (5,3), and the user’s heading angle was 3 degrees, which was small. On the other hand, the family or caregivers can lock or unlock the door for safety reasons. In our experiment, the door was unlocked, so the relay status was off. In addition, the family or caregivers can monitor these events and sensors’ status using the application; thus, they can allow the user to go outside or not.

### 3.3. Experiment C: Network Latency and System Response Time

In order to estimate the system’s complexity and performance, we evaluated the system’s computational response time and investigated the AR device’s battery usage when only fuzzy rule number 4 was activated. For instance, while the user’s position data were being transmitted to the message broker through the MQTT protocol, the response time for sending a voice message was measured.

In another experiment, we assessed the response time for demonstrating an AR image message after transmitting the new location of the user. Both tests were repeated under the same circumstances thirty times to estimate the accuracy of the system. Figure 11 shows the system’s response time for showing an AR message, such as an audio or picture message, in 30 different experiments. We first measured the response time for sending a voice message while the user moved around an object. Whenever the user was Near the object (at a distance of 1 to 5 dm), they received an audio message. We then repeated the same experiment thirty times. In some experiments, among these repetitions, the system’s response time for sending the audio message was around 500 ms, while it was higher in others (around 600 ms). However, most experiments had a response time with a value between 500 ms and 600 ms. Overall, the average response time for receiving an audio message was 553 ms.

In another experiment, we calculated the response time for sending an image message while the user moved around the same object. Whenever the user was Near the object (at a distance of 1 to 5 dm), they received an image message. We again repeated the same experiment thirty times. In some experiments, among these repetitions, the system’s response time for sending the image message was around 600 ms, while it was higher in others (approximately 750 ms). However, most experiments had a response time between 600 ms and 750 ms. Overall, the average response time for receiving an image message was 670 ms.

As we can see in Figure 11, in some experiments, the response time for sending audio and image messages is almost equal. This may occur due to the network latency, variation between the distance of the person and anchors, and different movement patterns in each type of experiment. We therefore discovered that playing a voice message rather than showing an AR image offers less battery consumption and higher performance. We also realized that the mean value of published-subscribed latency in the MQTT server is approximated to be 120 ms, and the data transmission rate is restricted, so the proposed algorithm is sufficiently precise for our application. Such time delays seem acceptable for real-time systems like our study. In another study [39], the authors estimated the response time in the form of an average response time value of 237.48 ms. They used INA 219 and PZEM004t as the sensors for IoT-based solar power plant monitoring without an additional decision-making process. They also utilized a communication protocol similar to ours, MQTT, but with ESP32 as a microcontroller, which is more potent than ESP8266.

## 4. Discussion

The AAL systems aim to develop innovation to keep people connected, healthy, active, and independent into their old age. They provide tools and services that improve the quality of life of those who face the challenges of aging and those who care for older people [40]. A survey of relevant literature shows the importance of developing assistive systems and monitoring services for older adults and caregivers [41,42,43,44,45,46,47,48]. Even though designing AAL systems receives some attention from researchers, this field still needs to be further developed, and there exist many essential issues in AAL systems and AR-based assistive tools.

Many research works have been focused on developing monitoring systems [41,49,50,51]. Monitoring older adults is essential; however, it is not sufficient for the end user to perform everyday tasks independently without any helping means. Some studies addressed the AR capabilities as a task-prompting system for assisting older adults [52,53]. In Ref [54], the authors implemented an assistive tool based on QR codes instead of an indoor positioning system without any decision-making algorithm to evaluate the system’s overall performance. Compared to our work, few such studies interrogate the technology potential for providing monitoring services and task-prompting systems as an integrated model. Most studies have addressed each problem separately and focused on designing a monitoring system or an assistive tool for older adults. Moreover, most of them are user-specific, non-scalable, and without any decision-making capability.

We employed ultra-wideband (UWB)-based location identification technology for in-home user tracking to improve the system’s scalability and accuracy. This scalable decision engine analyzes and determines data to make an appropriate decision based on fuzzy rules set. We proposed a comprehensive model to guide the development of such a system based on AR and IoT technologies and support its feasibility. The feasibility study helps develop integrative technological solutions by assessing the scalability and robustness of the technology, which seems to be missing in the current research works. Moreover, user participation in a feasibility study is essential to discover the end user’s expectations and wishes. Thus, we evaluated the system’s feasibility with user participation in the proposed scenarios. Moreover, we aimed at emerging sensor-based operations, which can be scalable for each user, adjusting for their preferences, which cannot be clearly seen in the other works, and most of them are user-specific.

More specifically, the system’s assistance relies on the effectual detection of ongoing activity. We presumed that the house has sufficient sensors and localization anchors to recognize the user’s current activity. Moreover, we took advantage of the experts’ discussion about people with MCI and their caregivers’ concerns [55] and defined a list of usage scenarios, experiments, and setups for our particular AAL system based on the proposed model as follows:Safety: One of the main challenges for the caregivers is to keep the person with memory impairment monitored. Caregivers cannot continuously watch the person, so an assistive system can benefit this case. The person can enter dangerous areas (for example, the balcony), fall, or even leave their home. In our experimental condition (Experiment and Setup A), we simulated this scenario and assessed the usability of the system in detecting dangerous areas and sending reminders to the user. The caregiver can also monitor these events.Personal assistance: Psychologists indicate that the engagement of people with memory impairment in routine activities is essential. An assistive tool with some functionalities of a personal assistant could be beneficial, mainly because people with MCI have problems with short-term memory and recent events. Thus, in Experiment and Setup B, we evaluated the system’s performance in sending correct reminders or turning off proper actuators to help the user complete an ongoing task based on the activated fuzzy rule. The system should perform correctly according to the object-based decision-making algorithm.Quick response: When people with memory impairment forget the place of an object or cannot complete an activity, they should be reminded how to make a correct decision. Otherwise, the person might be anxious or disappointed. In this regard, the AAL system should send reminders and alerts in real time. Thus, in the last experiment (Experiment and Setup C), we evaluated the system’s response time in sending such a message.

Indeed, these criteria and results can be helpful for future studies related to AAL systems’ development.

Moreover, the most critical challenge in developing AAL systems could be the acceptance of the end user. This challenge leads to requiring caregivers or family members to be able to take advantage of any application to support the person with memory impairment in their interaction with an assistive tool. The deviation of abilities, needs, and lifestyle varies from one person to another. Furthermore, users are typically not friendly to technical problems and do not show tolerance toward technical difficulties [3]. Thus, there should be a huge encouragement to participate in AAL systems. How to inspire people to take advantage of an assistance tool and trust in its privacy, security, and safety are considerable factors in the designing process of AAL systems [56].

On the other hand, different challenges in designing AR-based systems could be raised, which must be addressed in developing such systems [57,58]. Based on our results and evaluations, the following modifications can be suggested:The wearable devices, for example, localization tags and AR glasses, are required to be lightweight, small, consume lower power, and produce less heat, leading to scalability problems. As interesting as AR may look, some people may not be comfortable with wearing head-mounted displays all day. To overcome this challenge, we suggested lightweight AR glasses as an interaction device for the end user because of its lightweight and minimal heads-up screen, so that each person with MCI could potentially use it.In any AR application, it is necessary to be synchronized in real time for giving the user precise information. The device requires high bandwidth and the fastest data communication to keep the real-world and virtual content in sync. In this regard, the network latency in our proposed model should be satisfactory, and data loss should not occur after the execution of a series of data transfers.In some cases, AR can violate the user’s privacy and start saving personal preferences and information. In the proposed model, the collected data can only be shared with physicians or caregivers to monitor memory impairment progression and treatment response. Finally, individuals with MCI can control personal data based on their impairment stage.

## 5. Conclusions and Future Works

This work introduced a conceptual model to evaluate the possibility of integrating two emerging technologies—IoT and AR—for developing AAL. The implemented system offers AR functionality and features, including contextual information. The results identify that playing voice messages instead of showing images offers less battery consumption and higher performance. The results provide further evidence that the AAL system’s accuracy, reliability, and response time are appropriate for older people with memory impairment. The presented model can be scaled up to more sensors, actuators, scenarios, and rules and is customizable to improve the independence of the user. The simplicity of the proposed method supports the fact that users who are not accustomed to using new technologies, people experiencing disorientation, and, more generally, individuals with mild memory deficits could potentially benefit from the suggested system.

However, the validity of this study is restricted, as part of the work was implemented in a laboratory space. The presented scenarios are particular and must be precisely designed and considered for each person. Consequently, although the technical necessities are considered in our study, the specific context of the AR messages has to be customized for each person before the implementation of the whole system. The proposed system can add more fuzzy rules and be scalable for each user to achieve this goal. A novel AAL system can take advantage of all the relevant contextual dimensions that improve the proposed model. For example, it can extend the current model by considering location, time, objects, posture, frequency, and user activity history. Furthermore, physicians can potentially use the older adult’s in-home collected data to identify the progress of a specific disease or the effectiveness of the medication.

## Figures and Tables

**Figure 1 sensors-23-02673-f001:**
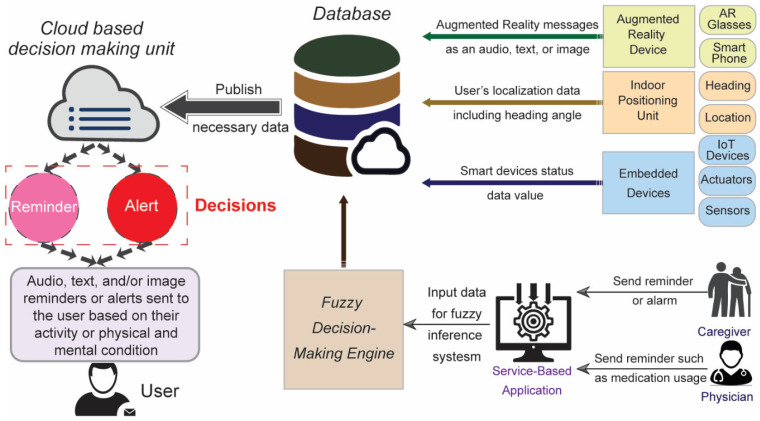
The proposed model for developing an AAL system. The model has four main components: (1) an indoor location and heading measurement unit in the local fog layer, (2) an AR application to make interactions with the end user, (3) a fuzzy decision-making system to handle the direct and environmental interactions with the user, and (4) a service-based application for caregivers or physicians to monitor the situation in real time and send reminders once required.

**Figure 2 sensors-23-02673-f002:**
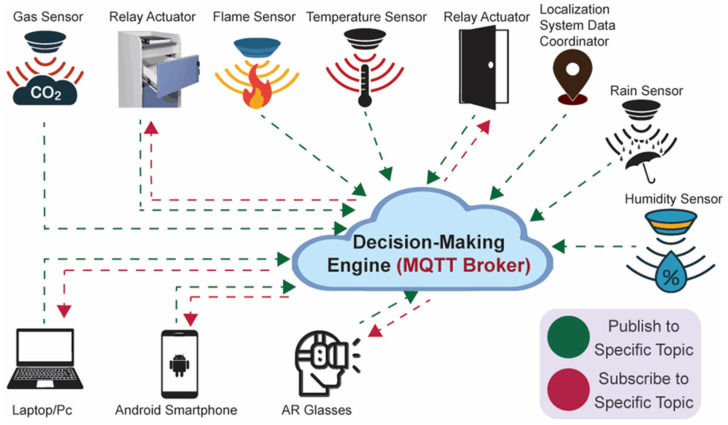
MQTT protocol operation in the AAL system. All the data and status of the smart devices will be transmitted/subscribed on a specific topic to the cloud. The user interaction within the AR environment can be rendered by AR glasses or mobile phone.

**Figure 3 sensors-23-02673-f003:**
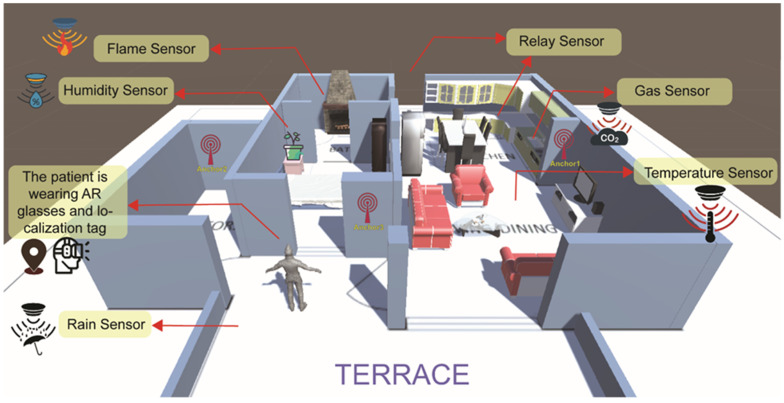
The general model of the AAL system in the older adult’s home environment is illustrated by the Unity game engine as an example. The location of IoT devices, indoor positioning anchors, and the user with an AR device is shown.

**Figure 4 sensors-23-02673-f004:**
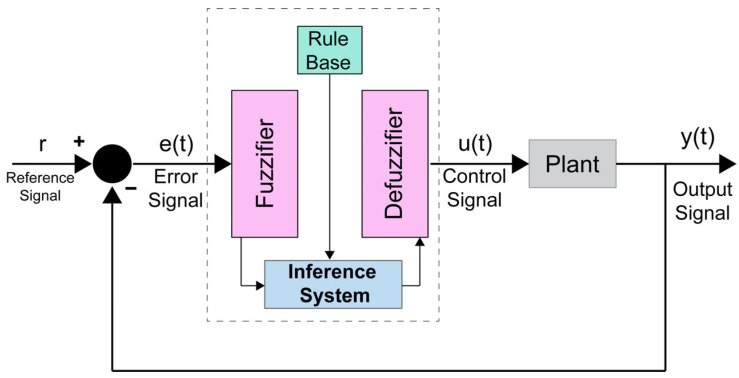
Fuzzy decision-making model. The fuzzy controller includes four elements: the inference system, rule base, defuzzifier interface, and fuzzifier interface. The output signal is the IoT device’s status or AR message values.

**Figure 5 sensors-23-02673-f005:**
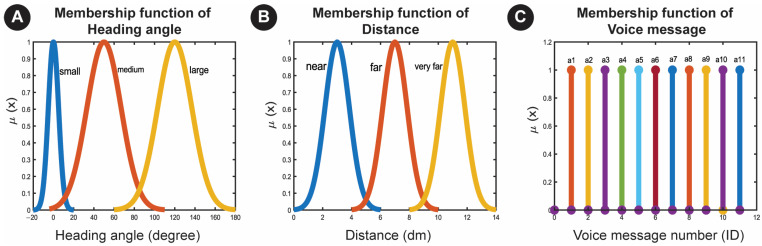
Membership function of (**A**) heading angle and (**B**) distance, which have Gaussian membership functions as a linguistic variable; (**C**) Membership functions of voice message (a: audio), which here has a fuzzy singleton membership function and is not a linguistic variable.

**Figure 6 sensors-23-02673-f006:**
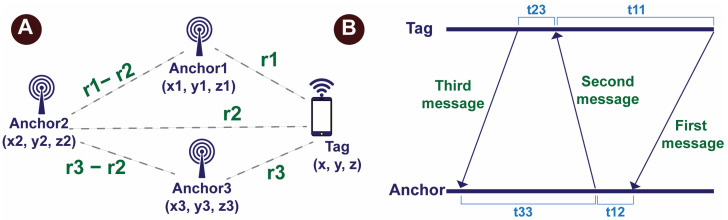
(**A**) TDoA method and (**B**) Round-trip time of arrival method. We can calculate the distance between predefined objects and the user according to the positioning tag and the location of three anchors in an indoor space.

**Figure 7 sensors-23-02673-f007:**
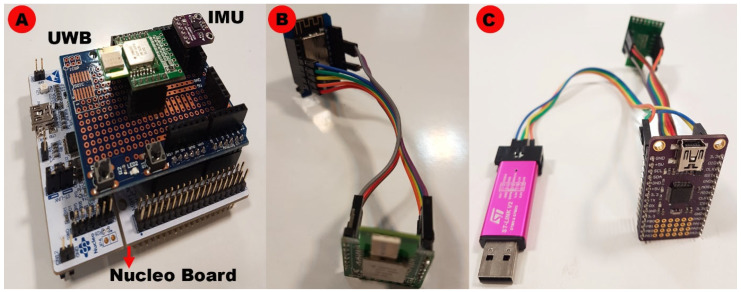
(**A**) Prototype used for user position estimation, (**B**) Data coordinator, (**C**) Anchor. The coordinator encapsulates and converges all the transmitted distances in the local fog for the final position estimation of the user.

**Figure 8 sensors-23-02673-f008:**
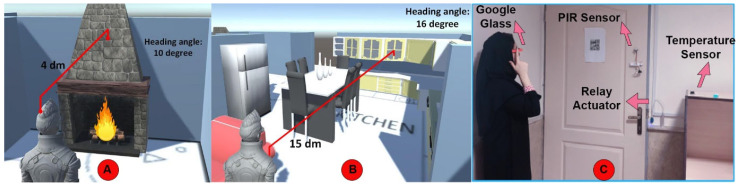
Illustration of the user interaction with home objects in (**A**) living room, (**B**) kitchen, and (**C**) experimental condition. The experimental condition was simulated on the Unity game engine to monitor the user’s indoor position and interaction with the smart objects.

**Figure 9 sensors-23-02673-f009:**
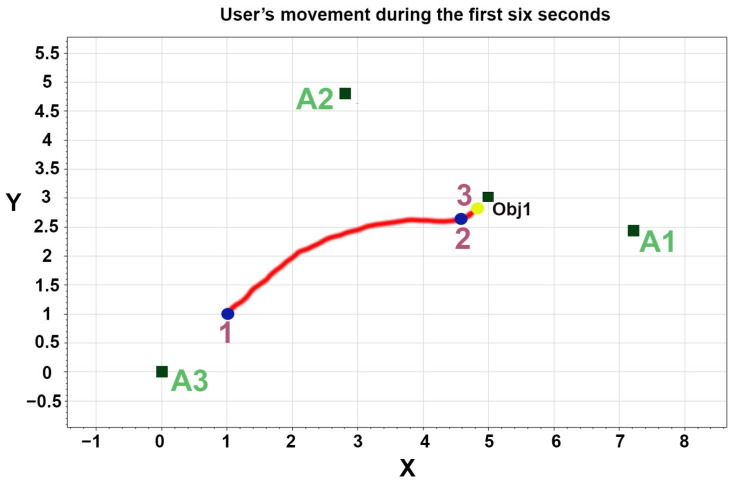
The three anchors of the indoor positioning system’s location (A1, A2, A3). The object location was shown as (Obj1). The user’s path during the first six seconds of Experiment B was also illustrated (starting from points 1 to 3). The final point of the user: (4.8,2.8), heading angle: 3 degrees.

**Figure 10 sensors-23-02673-f010:**
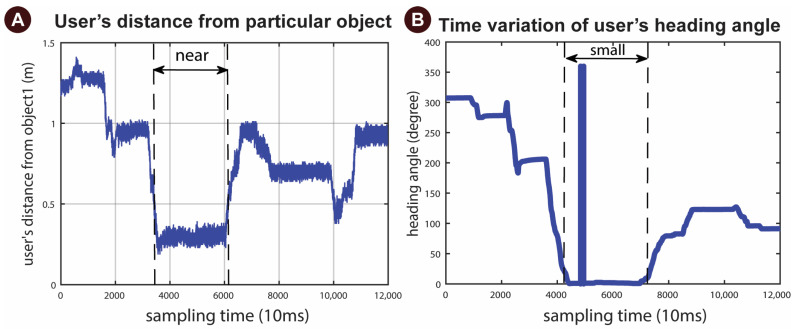
Result of (**A**) the time variation of the user’s distance from a particular object during the experiment, (**B**) the pattern of the time variation of the user’s heading angle during Experiment B. When the heading and distance values are members of the small fuzzy member functions, the corresponding fuzzy rule will be activated.

**Figure 11 sensors-23-02673-f011:**
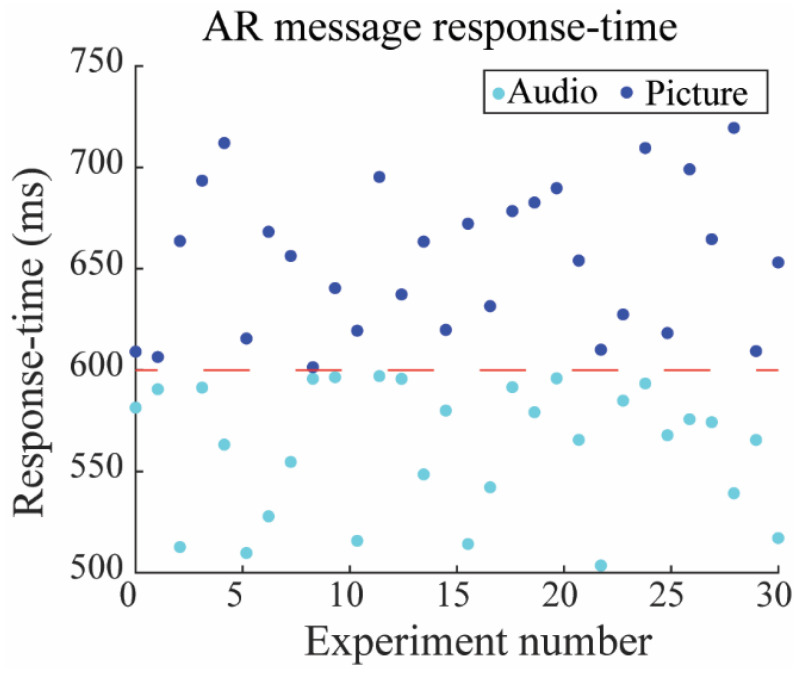
AR audio and picture message response time after the decision-making process. An average of 553 ms is required to play a voice message, and to show a three-dimensional image message, the mean value of response time is 670 ms.

**Table 1 sensors-23-02673-t001:** Explanation of outputs and inputs of fuzzy decision-making unit. The fuzzy membership functions of each variable can be discrete or have a Gaussian distribution according to their data type.

Parameter	Fuzzy Membership Functions	Data Type	Direction
Heading angle	Large, medium, and small	Linguistic	Input
Distance	Very far, far, near	Linguistic	Input
Time	Midnight night evening, late afternoon early afternoon morning, early morning	Linguistic	Input
Temperature	Very hot, hot, warm, mild, cool, cold, very cold	Linguistic	Input
Humidity	Very humid, humid, dry, very dry	Linguistic	Input
Rain detection	Yes, No	Boolean	Input
Flame detection	Yes, No	Boolean	Input
Gas detection	Yes, No	Boolean	Input
Relay status	Yes, No	Boolean	Output
Voice message	1, 2, …, 20	Integer	Output
Image message	1, 2, …, 20	Integer	Output
Text message	1, 2, …, 10	Integer	Output

**Table 2 sensors-23-02673-t002:** Result of the experimental test. There were 20 experiments with various variations on the user’s localization data and IoT devices’ status. There were five different objects, and the user’s distance from them was measured simultaneously. The localization data can be varied through the user’s movement and heading angle. We also checked the activated fuzzy rule and its reliability in each experiment.

Object Number	Experiment Number	Distance (dm)	Heading Angle (degree)	Membership Function of Distance	Membership Function of Heading Angle	Activated Rule Number
1	1 2 3 4	3 4 8 14	12 35 10 16	Near Near Far -	Small Medium Small Small	12 - - -
2	5 6 7 8	10 7 12 2	42 12 73 11	Very Far Far Very Far Near	Medium Small Large Small	- - - 14
3	9 10 11 12	5 16 6 13	122 9 38 12	Near - Far Very Far	- Small Medium Small	11 - - -
4	13 14 15 16	18 4 11 7	0 10 14 72	- Near Very Far Far	Small Small Small Large	- 21 - -
5	17 18 19 20	12 2 17 13	86 12 8 76	Very Far Near - Very Far	Large Small Small Large	- 2 - -

**Table 3 sensors-23-02673-t003:** Experimental condition in experiment B. We evaluated the indoor positioning system’s accuracy in estimating users’ positioning data and the distance from a specific fixed object.

Experiment Number	User’s Location (x,y)	User’s Heading Angle (degree)	Location of Anchor1(A1) (x,y)	Location of Anchor2(A2) (x,y)	Location of Anchor3(A3) (x,y)	Location of an Object (x,y)
1	(1,1)	263	(0,0)	(2.8,4.8)	(7.2,2.4)	(5,3)
2	(4.6,2.6)	105	(0,0)	(2.8,4.8)	(7.2,2.4)	(5,3)
3	(4.8,2.8)	3	(0,0)	(2.8,4.8)	(7.2,2.4)	(5,3)

Measurement unit: meter, room dimensions: 8.5 × 4.6 × 2, object (Obj1) dimensions: 0.94 × 0.08 × 1.9. Rain sensor’s location: (3,4.5).

## Data Availability

This study’s data are available from the corresponding author following a reasonable request. The customized software and applications designed in the present study are available from the corresponding author following a reasonable request.

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
