# Peer review of "A Decision-Aware Ambient Assisted Living System with IoT Embedded Device for In-Home Monitoring of Older Adults"

_sensors, 2023, doi:10.3390/s23052673_

Round 1

Reviewer 1 Report

Minor comments:

1. Spell out first use of all acronyms. I realize this is a specialized field but the acronyms still need to be spelled out every first reference. 

2. The reference on line 36 is from 2012--need a more recent reference if stating that the current main problem is MCI/loss of memory.

3. Not sure what is meant by use of open boxes in line 42. please clarify. 

4. Explain what AR-based serious games are in line 48.

5. State what cloud-based fuzzy decision-making engines are. Line 88.

6. Need citation on line 93-94 on what these past experiments have been.

7. Fix typo on line 127, I think it is supposed to say "The end-user puts on".

8. Spell out MQTT.

9. Need a citation for sentence 552-553 that users are not friendly to tech problems and don't show tolerance. 

Major Comments:

1. Rather than use of "elderly" the term widely accepted in the literature is "older adult". Change all instances of "elderly" to "older adults."

2. Introduction is confusing, jumping to discussing caregivers in second paragraph. please clarify.

3. Unclear if the Internet of Things are apps on tablets, smart phones, or somewhere else. Clarify in paragraph that begins with line 64.

4. State who has the funds or the means to have Ambient Assisted Living (AAL) and IoT apps/ devices. Seems to be a divide here in who may be able to afford these. If the main aim of the paper is to provide a conceptual model of emerging tech to design AAL systems this needs to be clear for readers. Also need better description of what AAL are in materials and methods section. 

5. Good outline starting line 99 on paper organization. 

6. Describe to what server and what database IoT data will be published. Lines 120-122

7. Unclear what 2.1 d) means. Especially second sentence. 

8. In Figure 1 unclear what local fog layer and fuzzy decision making system mean. State what these are. 

9. I am not an expert on system architecture. Please rely on other reviewers for feedback here. Same with input/output variables. I will say I had a hard time following this, so if you are targeting a wider audience, you will want to explain more of the technical aspects for your readers who are unfamiliar with them.

10. Sentence on 516-517 is confusing as written. Says that the authors' tried to individualize their AAL technology for each user and that this can't be seen in other works, which are user-specific. But the authors just stated their work was individualized for each user. Please clarify. 

11. Discussion mentions Google Glass which I believe was discontinued. Authors should check if Google is still developing these. If not need to adjust discussion. 

12. Overall this is important work but I found the paper hard to follow overall and with way too much information. Suggest dividing it in to two papers: one for the conceptual model and sample scenarios, and the other explaining system architecture. It is a lot to digest in one reading.  

Author Response

We would like to thank reviewer 1 for their valuable comments. We greatly appreciate their comments and suggestions, which helped us to revise and improve our paper, in addition to being an effective guiding principle for our research. The comments have been carefully studied, and corrections have been made, which we hope will be accepted. During the revision of the paper, blue highlights are used to indicate what revisions have been made. Below, we answer each of the points raised.

Reviewer 2 Report

The paper is well written and has many important experimental results.

However, the authors should be clearer on the use of the fuzzy decision.

As readers, I can't see what will happen without using them!

********

Line 312:  details about the used prototype?

In the paper, I don’t see a complete description of used sensors.

Equation (1) :   M(x,y,z)

Equation (2) :  ri2  = (xi - x)2 + …

Table 2:  Give more details and explications for all experiment scenario’s.  

The activated rule for 2, 3, 4, 5, … Experiments ?

Line 393: Some other experiments have not activated any fuzzy rules because of the input values and membership functions? More arguments

Line 434:  When the distance value lies within 1 to 5 dm, it
becomes a member of the near fuzzy membership function? More explication about these!

Line 475:  about response time: have you some reference values to compare with that found in this topic?

What about human and older adult’s reaction?

Figure 11:  image message received by the user … note necessary in the paper. No added value for publication!

Figure 12:  The response time of some picture and audio messages is the same and around 600 ms ? For the same experimental number!  

*******

It will be more significant if you added some previous works and to compare with.  Maybe an added value for your work.

In general, you have done a good job and in my opinion can be presented with more added details … (what is the impact of using fuzzy decision?)

Author Response

We are grateful for the valuable comments provided by reviewer 2. Your comments and suggestions greatly helped us revise and improve our paper. We have carefully studied the comments and made corrections, which we hope will be accepted. This paper has been revised during revision, as indicated by blue highlights. Here are our responses to each of your points.

Reviewer 3 Report

1.The block diagram in Figure 4 has some errors in the direction of the arrows, which need to be corrected.

2. Page 7, line 228, "The triangular membership function defines the relay actuators' ....", seems to be wrong for this description.

3. The description of the fuzzy rule base on page 7 seems to be too simple to see the relationship between the former part and the latter part of the If Then rule.

Author Response

Please accept our sincere thanks for reviewer 3's helpful comments. The comments have been carefully reviewed, and corrections have been made. Revisions have been made to this paper, as indicated by blue highlights. We have responded to each of your points below.
